# Tofacitinib May Inhibit Myofibroblast Differentiation from Rheumatoid-Fibroblast-like Synoviocytes Induced by TGF-β and IL-6

**DOI:** 10.3390/ph15050622

**Published:** 2022-05-18

**Authors:** Piero Ruscitti, Vasiliki Liakouli, Noemi Panzera, Adriano Angelucci, Onorina Berardicurti, Elena Di Nino, Luca Navarini, Marta Vomero, Francesco Ursini, Daniele Mauro, Vincenza Dolo, Francesco Ciccia, Roberto Giacomelli, Paola Cipriani

**Affiliations:** 1Department of Biotechnological and Applied Clinical Sciences, University of L’Aquila, 67100 L’Aquila, Italy; noemi.panzera@graduate.univaq.it (N.P.); adriano.angelucci@univaq.it (A.A.); onorina.berardicurti@graduate.univaq.it (O.B.); elena.dinino@student.univaq.it (E.D.N.); paola.cipriani@univaq.it (P.C.); 2Rheumatology Section, Department of Precision Medicine, University of Campania “Luigi Vanvitelli”, 80131 Naples, Italy; vasiliki.liakouli@unicampania.it (V.L.); daniele.mauro@unicampania.it (D.M.); francesco.ciccia@unicampania.it (F.C.); 3Rheumatology, Immunology, and Clinical Medicine Research Unit, Department of Medicine, Campus Bio-Medico University of Rome, 00128 Rome, Italy; l.navarini@unicampus.it (L.N.); m.vomero@unicampus.it (M.V.); r.giacomelli@unicampus.it (R.G.); 4Immunorheumatology Unit, Fondazione Policlinico Universitario Campus Bio-Medico, 00128 Rome, Italy; 5Medicine and Rheumatology Unit, IRCCS Istituto Ortopedico Rizzoli, 40136 Bologna, Italy; francesco.ursini2@unibo.it; 6Department of Biomedical and Neuromotor Sciences (DIBINEM), Alma Mater Studiorum University of Bologna, 40126 Bologna, Italy; 7Department of Life, Health and Environmental Sciences, University of L’Aquila, 67100 L’Aquila, Italy; vincenza.dolo@univaq.it

**Keywords:** rheumatoid arthritis, myofibroblast, tofacitinib

## Abstract

During rheumatoid arthritis (RA), the pathogenic role of resident cells within the synovial membrane is suggested, especially for a population frequently referred to as fibroblast-like synoviocytes (FLSs). In this study, we assess the markers of myofibroblast differentiation of RA-FLSs by ex vivo observations and in vitro evaluations following the stimulation with both TGF-β and IL-6. Furthermore, we investigated the possible inhibiting role of tofacitinib, a JAK inhibitor, in this context. Myofibroblast differentiation markers were evaluated on RA synovial tissues by immune-fluorescence or immune-histochemistry. RA-FLSs, stimulated with transforming growth factor (TGF-β) and interleukin-6 (IL-6) with/without tofacitinib, were assessed for myofibroblast differentiation markers expression by qRT-PCR and Western blot. The same markers were evaluated following JAK-1 silencing by siRNA assay. The presence of myofibroblast differentiation markers in RA synovial tissue was significantly higher than healthy controls. Ex vivo, α-SMA was increased, whereas E-Cadherin decreased. In vitro, TGF-β and IL-6 stimulation of RA-FLSs promoted a significant increased mRNA expression of collagen I and α-SMA, whereas E-Cadherin mRNA expression was decreased. In the same conditions, the stimulation with tofacitinib significantly reduced the mRNA expression of collagen I and α-SMA, even if the Western blot did not confirm this finding. JAK-1 gene silencing did not fully prevent the effects of stimulation with TGF-β and IL-6 on these features. TGF-β and IL-6 stimulation may play a role in mediating myofibroblast differentiation from RA-FLSs, promoting collagen I and α-SMA while decreasing E-Cadherin. Following the same stimulation, tofacitinib reduced the increases of both collagen I and α-SMA on RA-FLSs, although further studies are needed to fully evaluate this issue and confirm our results.

## 1. Introduction

Rheumatoid arthritis (RA) is a chronic systemic inflammatory disease mainly affecting the joints, characterized by synovial hyperplasia and inflammatory cells infiltration [1]. During the disease, the pathogenic role of resident cells within the synovial membrane is suggested, especially for a population frequently referred to as fibroblast-like synoviocytes (FLSs) [2]. These cells are accumulated to form a distinct structure called the synovial lining layer and express mesenchymal markers common to fibroblasts [3]. During RA, FLSs may contribute to the inflammatory process within synovial tissues, culminating in rheumatoid pannus formation [1,2,3]. Further, the synovial lining is described as a mesenchymal tissue because its function and morphology may resemble those of epithelial tissues [4,5]. The changes, which occur in the synovial lining during development of RA, may mirror a process called epithelial-to-mesenchymal transition (EMT) [5]. The latter implies that polarized epithelial cells may undergo sequential changes toward a mesenchymal cell phenotype with an increased expression of α-smooth muscle actin (α-SMA), collagen type I (collagen I), vimentin as well as a decreased expression of E-cadherin (E-Cad) [6]. By both EMT and the activation of fibroblasts, myofibroblasts are formed and are responsible for the overproduction of extracellular matrix during fibrosis [6,7]. The accumulation of extracellular matrix is also found in RA, indicating that both the gain of invasiveness of RA-FLSs and the increased matrix production observed in fibrosis may represent an additional pathophysiological event mediated by synoviocytes [5,6,7]. In this context, the role of transforming growth factor-β (TGF-β) and interleukin (IL)-6 in inducing the migration and invasion of RA-FLSs has been recently proposed in inducing an activated phenotype of these cells [8,9]. Interestingly, the JAK/STAT (Janus kinase and signal transducer and activators of transcription) pathway may be implicated for TGF-β- and IL-6-induced effects on RA-FLSs [10,11]. The four JAK proteins (JAK1, 2, 3, and Tyk2) are a family of signaling molecules that are associated with cytokine receptors [12]. Combinations of JAK1 and/or JAK2 transduce signals when members of the IL-6 family ligate their receptors, thus activating their downstream targets, such as STAT1 and STAT3 [12,13]. JAK1/JAK3 combinations induce signaling for cytokines that use the common gamma-chain receptor, particularly T cell cytokines [14,15]. Tofacitinib is a targeted small molecule inhibitor of several JAK isoforms, especially JAK3 and JAK1, and its inhibition of signal transduction may influence the cellular response to the inflammatory environment [12,13,14,15,16]. As observed in other diseases [16,17,18,19,20,21,22], tofacitinib could also exert its therapeutic effects via an inhibition of myofibroblast differentiation from RA-FLSs, although it has not been fully elucidated yet. On these bases, in this study, we aimed to assess the markers of myofibroblast differentiation of RA-FLSs by ex vivo observations and in vitro evaluations following the stimulation with both TGF-β and IL-6. Furthermore, we investigated the possible inhibiting role of tofacitinib in this context.

## 2. Results

### 2.1. JAK-1, STAT-3, and Markers of Myofibroblast Differentiation in RA Synovial Tissues

JAK-1 and STAT-3 were significantly expressed in the synovial lining layer and in the vascular cells of the RA synovial samples (JAK-1, *p* = 0.0006; STAT-3, *p* = 0.0003) (Appendix A). As reported in Figure 1, α-SMA was poorly expressed in the HC synovial tissues, whereas the cells represented in the synovial lining layer expressed high levels of E-Cad (Figure 1A). On the contrary, in the RA synovial tissues (Figure 1B), α-SMA was increased and E-Cad was more significantly decreased than HCs (α-SMA *p* = 0.01, E-Cad *p* = 0.001, respectively). Furthermore, vimentin was expressed in HC synovial tissues, but the intensity of expression was not significantly different than RA.

### 2.2. TGF-β and IL-6 Promoted Myofibroblast Differentiation in RA-FLSs

As reported in Figure 2A, following the stimulation with TGF-β + IL-6 for 6 days, the collagen I mRNA expression was significantly increased in both RA-FLSs (*p* = 0.002) and HC-FLSs (*p* = 0.002) more than the UT-cells. Interestingly, the effects of TGF-β + IL-6 on the collagen I expression in RA-FLSs were significantly higher than HCs-FLSs (*p* = 0.002). Furthermore, following TGF-β + IL-6 stimulation, the E-Cad mRNA expression was significantly decreased in both RA-FLSs (*p* = 0.002) and HC-FLSs (*p* = 0.002), without significant differences between HC- and RA-FLSs (Figure 2B). Conversely, TGF-β + IL-6 did not induce a significant change in the levels of expression of vimentin mRNA in both HCs- and RA-FLSs (Figure 2C).

Concerning α-SMA mRNA expression (Figure 2D), only in RA-FLSs, TGF-β + IL-6 induced a significant increased expression more than UT-RA-FLSs. Mirroring qRT-PCR gene expressions, following TGF-β + IL-6, the collagen I (*p* = 0.049) and α-SMA (*p* = 0.049) protein expressions were significantly increased in RA-FLSs more than UT-cells (Figure 2E and Appendix A).

In the same conditions, the stimulation with tofacitinib significantly reduced the mRNA expression of both collagen I (Figure 2A, *p* = 0.01) and α-SMA (Figure 2D, *p* = 0.004) more than TGF-β + IL-6 treated cells, even if the Western blot did not confirm this finding (Appendix A). TGF-β + IL-6 stimulation induced a significant increase in migrated RA-FLSs more than UT cells (*p* = 0.0004). Tofacitinib did not fully prevent the migration of RA-FLSs (Appendix A).

### 2.3. JAK-1 Gene Silencing Did Not Fully Affect Myofibroblast Differentiation in RA-FLSs

To address the role of JAK-1 during the induction of myofibroblast markers in RA-FLSs treated with TGF-β + IL-6, we silenced the JAK-1 gene using JAK-1-siRNA or scr-siRNA. JAK-1-siRNA efficiently knocked down the JAK-1 molecule in RA-FLSs, and, after silencing, TGF-β + IL-6 did not modulate the JAK-1 expression (*p* = 0.002) (Figure 3A). Interestingly, in cells transfected with scr-siRNA, TGF-β + IL-6 stimulation induced a significant decrease in the JAK-1 gene expression (*p* = 0.002) (Figure 3A). Furthermore, in RA-FLSs silenced with JAK-1-siRNA, TGF-β + IL-6 induced a significant increase in collagen I mRNA expression more than UT cells (*p* = 0.002) (Figure 3B). These results paralleled with those derived from the cells transfected with scr-siRNA (*p* = 0.002). Additionally, in cells silenced with JAK-1-siRNA, TGF-β + IL-6 promoted a significant decrease in E-Cad mRNA expression (*p* = 0.004) more than UT cells, paralleling the results obtained in cells transfected with scr-siRNA (*p* = 0.004) (Figure 3C). Furthermore, the silencing of JAK-1 in UT cells induced a significant decrease in E-Cad mRNA expression more than UT cells transfected with scr-siRNA (*p* = 0.002). Additionally, in scr-siRNA, TGF-β + IL-6 induced an increase in α-SMA mRNA levels (*p* = 0.04). However, the silencing of JAK-1 made TGF-β + IL-6 unable to induce an increase in α-SMA mRNA expression.

## 3. Discussion

In this work, the stimulation of RA-FLS with both TGF-β and IL-6 down-regulated the expression of E-cadherin and simultaneously up-regulated those of both collagen I and α-SMA in promoting an activated phenotype on these cells. Tofacitinib significantly reduced the mRNA expression of both collagen I and α-SMA on RA-FLSs, advocating a possible further mechanism of action of the drug in RA, although additional studies are needed to fully evaluate this issue and confirm our results.

In RA synovial tissues, α-SMA was increased, and, simultaneously, E-Cad was decreased more than HCs, suggesting the presence of markers associated with myofibroblast differentiation [7]. To further investigate this issue, we isolated RA-FLSs and these cells were stimulated with both TGF-β and IL-6. A down-regulation of E-cad and a concomitant up-regulation of both collagen I and α-SMA were observed, suggesting the induction of a more invasive phenotype on RA-FLSs. In fact, TGF-β and IL-6 decreased E-Cad, which is an important component of extracellular connections, and its down-regulation may facilitate RA-FLSs migration and invasion [5,20]. Furthermore, TGF-β and IL-6 increased collagen I and α-SMA, which are markers of myofibroblast differentiation and activation [7,16,18,19]. Thus, an activated phenotype of RA-FLSs may be induced by these molecules, supporting the hypothesis that a process of EMT may be involved in the pathogenesis of the disease [23,24]. EMT could contribute to pannus formation and, ultimately, to joint destruction in RA [5,23,24]. Considering the role of the JAK/STAT pathway in this context [10,11], we assessed the possible inhibiting role of tofacitinib. Based on our results, tofacitinib could inhibit this mechanism since it reduced the mRNA expression of both collagen I and α-SMA in RA-FLSs following the stimulation with both TGF-β and IL-6. However, the Western blot analysis did not fully confirm this finding, highlighting the need of further studies to entirely clarify this topic. Probably, a longer time of stimulation with tofacitinib might confirm the results of the RT-PCR with the Western blot analysis as well [17,18,19].

In addition, JAK-1 gene silencing did not fully prevent the effects of stimulation with TGF-β and IL-6 on these features. Conflicting results are available regarding this topic, probably related to the different experimental conditions [25,26,27,28]. TGF-β and IL-6 may regulate the JAK/STAT pathway either in a positive or negative manner, depending on the cell type [25,26,27,28], and further studies are required to entirely elucidate these issues.

Our work is affected by some limitations; therefore, our results should be carefully interpreted and subsequent confirmatory studies are warranted. We did not evaluate the phosphorylation of JAK/STAT factors in our cell cultures following the administration of tofacitinib. However, the inhibition of the phosphorylation of JAK/STAT factors on synovial cells as well as on immune cells following the treatment with tofacitinib has already been demonstrated [29,30,31]. On these bases, we focused our work on further possible mechanisms of tofacitinib in RA-FLSs, exploring additional effects of this drug on markers of fibroblast activation. Thus, further studies are needed to fully confirm these findings.

In conclusion, the stimulation of RA-FLS with both TGF-β and IL-6 down-regulated the expression of E-cadherin and simultaneously up-regulated those of both collagen I and α-SMA in promoting an activated phenotype on these cells. Although further studies are needed to confirm our findings, tofacitinib reduced the expression of both collagen I and α-SMA on RA-FLSs, suggesting an additional mechanism of action of this drug. Finally, JAK-1 gene silencing did not fully affect myofibroblast differentiation in RA-FLSs, suggesting that other additional mechanisms could be involved in this process.

## 4. Materials and Methods

### 4.1. Patients and Samples Collection

The synovial tissues were obtained from involved knees of 7 patients affected by moderate to active RA (DAS28 > 3.2) who fulfilled 2010 ACR/EULAR classification criteria [32] (Appendix A) but who were not treated with biologic DMARDs. Normal synovial tissues were obtained from 7 healthy controls (HCs) who underwent surgery due to knee trauma. The local ethics committee approved the study protocol (ASL1, Avezzano-Sulmona-L’Aquila, L’Aquila, Italy, protocol number #11261) and it has been performed according to the Good Clinical Practice guidelines and the Declaration of Helsinki. Informed consent was obtained from all involved participants.

### 4.2. Reagents

Human recombinant TGF-β was obtained from R&D Systems Inc. (Minneapolis, MN, USA). We used TGFb1 isoform that was diluted in sterile 4 mM HCl containing 1 mg/mL human serum albumin (HSA). IL-6 was obtained from Peprotech (East Windsor, NJ, USA) and was reconstituted in sterile PBS containing 0.1% HSA.

### 4.3. Immunofluorescence

Synovial sections (thickness 3 µm), fixed in paraffin, were deparaffinized and treated with protein block (DAKO, Santa Clara, CA, USA). After blocking, sections were incubated with primary antibodies, including conjugated anti-α-SMA (Sigma-Aldrich, Burlington, MA, USA), anti-E-Cad (Proteintech, Chicago, IL, USA), and anti-vimentin (Proteintech, USA). Visualization of the primary antibodies was performed using Alexa Fluor 488-conjugated (Invitrogen, Waltham, MA, USA). Negative controls were obtained by omitting the primary antibody. Sections were examined and photographed with Olympus BX53 fluorescence microscope. The intensity of fluorescence was measured by using NIHimageJ version 1.5 freeware.

### 4.4. Immunohistochemistry

Synovial sections (thickness 3 µm), fixed in paraffin, were deparaffinized and treated with peroxidase-blocking reagent (DAKO, USA) to inactivate endogenous peroxidase and then with protein block (DAKO, USA) to block non-specific binding. After blocking, sections were incubated with primary antibodies, including anti-JAK-1 (Abcam, Cambridge, UK), and anti-STAT-3 (Abcam, UK). Visualizations of the primary antibodies were performed using DAB (diaminobenzidine) (DAKO, USA). Negative controls were obtained by omitting the primary antibody. Sections were examined and photographed under light microscope (Olympus BX53). The OD was measured by using NIHimageJ version 1.5 freeware.

### 4.5. FLSs Isolation and Culture

Synovial fibroblasts were obtained from the synovium of RA patients and HCs. Synovium was minced and incubated with 1 mg/mL collagenase type VIII (Sigma-Aldrich, USA) in serum-free RPMI-1640 medium (Life Technologies, Carlsbad, CA, USA) for 1 h at 37 °C. Successively, the digested synovium was filtered, washed extensively, and cultured in Dulbecco’s modified Eagle’s medium (DMEM) medium (Sigma, St. Louis, MO, USA) supplemented with 10% FBS (Standard South America origin, Lonza Cohasset, MN, USA) and 100 units/mL penicillin, and 100 ng/mL streptomycin (Sigma, USA) at 37 °C in a humidified atmosphere of 5% CO_2_. FLSs were used from passages 3 to 7. Isolated FLSs were characterized by flow cytometry (Becton Dickinson FACS Melody cell sorter and Becton Dickinson FACS Chorus™ software) as a homogeneous population (phenotype: 0% CD14+, >90% CD90+ and 7% CD68+/CD90+). CD90+ (Thy1.1), CD14+ and, CD68+ cells were gated from the whole obtained population and stained according to manufacturer instructions (ThermoFischer Scientific, Rodano, Italy).

### 4.6. FLSs stimulation with TGF-β + IL-6

FLSs were stimulated with both TGF-β 10 ng/mL (R&D, USA) and IL-6 100 ng/mL (Proteintech, USA) according to previous works [8,10]. To establish the optimal timing for these two molecules in our system, we performed a pilot experiment assessing the α-SMA mRNA expression (data not shown) using P3 FLSs obtained from one patient, stimulated with TGF-β and IL-6, alone and together, for 2 days and 6 days. The best retrieved condition for FLSs stimulation was TGF-β and IL-6 together for 6 days (these molecules were administered every 2 days).

### 4.7. FLSs Stimulation with Tofacitinib

In addition to TGF-β + IL-6, FLSs were also treated with tofacitinib (1 μM) to evaluate its inhibitory effects on expression of myofibroblast differentiation. The concentration of tofacitinib followed what was already published in similar experimental models [14,15].

### 4.8. qRT-PCR Analysisc

Total RNA was extracted from FLSs using All prep DNA/RNA/miRNA universal kit (Qiagen, Hilden, Germany) and reverse transcribed into complementary DNA (cDNA) with the High Capacity cDNA Reverse transcription kit (Applied Biosystems, Waltham, MA, USA). The qRT-PCR was performed by using SYBR green kits (Applied Biosystems, Bleiswijk, The Netherlands). The qRT-PCR was run in triplicate. Primers were designed based on the reported sequences (Primer bank NCBI: β-Actin: 5′-CCTGGCACCCAGCACAAT-3′(forward) and 5′-AGTACTCCGTGTGGATCGGC-3′(reverse); α-SMA: 5′-CGGTGCTGTCTCTCTATGCC-3′(forward) and 5′-CGCTCAGTCAGGATCTTCA-3′(reverse); collagen I: 5′-AGGGCCAAGACGAAGACA-3′(forward) and 5′-AGATCACGTCATCGCACAACA-3′(reverse); vimentin: 5′-AGTCCACTGAGTACCGGAGAC-3′(forward) and 5′-CATTTCACGCATCTGGCGTTC-3′(reverse); E-Cad: 5′-CGA-GAGCTACACGTTCACGG-3′(forward) and 5′-GGGTGTCGAGGGAAAAA-TAGG-3′(reverse)). Results were analyzed after 45 cycles of amplification using the ABI 7500 Fast Real Time PCR System.

### 4.9. Western Blot

FLSs were lysed in lysis buffer (RIPA buffer, Cell Signaling Danvers, MA, USA) for 10 min and cleared by centrifugation. The protein concentration was calculated by Bicinchoninic Protein Assay kit (EuroClone, Pero, Italy). Forty µg of proteins was separated by SDS-PAGE and transferred to nitrocellulose membranes. After blocking in 10% non-fat milk in tris-buffered saline/1% tween 20 (TBS/T), as previously reported [16], incubation took place with the following primary antibodies collagen I (ThermoFisher Scientific, Waltham, MA, USA), α-SMA (Abcam, UK), vimentin (Proteintech, USA), E-Cad (Proteintech, USA), and CDH-11 (ThermoFisher Scientific, USA). Successively, horseradish peroxidase-conjugated secondary antibodies (Cell Signaling, Danvers, MA, USA) were appropriately used. The detection was performed by Long Lasting Chemiluminescent Substrate (EuroClone, Pero, Italy). All the signals were quantified by normalizing against a β-actin signal (Santa Cruz Biotechnology, Dallas, TX, USA). Immunoreactive bands were acquired by chemidoc (ImageLab). All the bands were quantified by densitometry using NIHimageJ version 1.5 freeware. The experiments were performed in triplicate, and the blot of each experiment was reported in supporting information file.

### 4.10. Chemoinvasion Assays

FLSs chemoinvasion was evaluated by 48-well modified Boyden chamber. We used filters (8 mm) coated with Matrigel. For the evaluation of the basal motility, medium supplemented with 0.5% FBS was used in the lower chamber. Following the stimulation with both TGF-β 10 ng/mL and IL-6 100 ng/mL and/or tofacitinib (1 µM), the cells were added to the upper chamber at a density of 5 × 10^4^ cells per well and suspended in media containing 2% fetal bovine serum. After 12 h of incubation at 37 °C, non-migrated cells on the upper surface of the filter were removed by scraping. The cells that migrated to the lower side of the filter were stained with Diff-Quik stain and counted using an Olympus BX53 microscope. The assays were run in triplicate. Results were reported as median (range) of number of cells migrated per microscopic field.

### 4.11. siRNA Assay

To silence JAK-1 expression, RA-FLSs were transfected with Silencer Select JAK-1-siRNA (Life Technologies, USA) or with Silencer Select non-targeting scramble siRNA (scr) (Life Technologies, USA) using Lipofectamine™ 2000 (Life Technologies, USA). Transfection was performed according to the manufacturer’s instructions. Briefly, RA-FLSs were plated in completed medium 24 h prior to transfection. After, the cultures were incubated for 24 h with 30 pmol of siRNA in 2 mL of OptiMem. After incubation, plates were washed, and cells were allowed to recover in growth conditions (1% FBS) supplemented with TGF-β (10 ng/mL) and IL-6 (100 ng/mL) for 6 days (the administration was repeated every 2 days).

### 4.12. Statistical Analysis

GraphPad Prism 5.0 software was used for statistical analyses. Results were expressed as median (range) due to the non-parametric distribution of our data. The Mann–Whitney U test was used as appropriate for analyses. Statistical significance was expressed by a *p* value ≤ 0.05.

## 5. Conclusions

The stimulation of RA-FLS with both TGF-β and IL-6 down-regulated the expression of E-cadherin and simultaneously up-regulated the expression of both collagen I and α-SMA in promoting an activated phenotype on these cells. Although further studies are needed to confirm our findings, tofacitinib reduced the expression of both collagen I and α-SMA on RA-FLSs, suggesting an additional mechanism of action of this drug. Finally, JAK-1 gene silencing did not fully affect myofibroblast differentiation in RA-FLSs, suggesting that other additional mechanisms could be involved in this process.

## Figures and Tables

**Figure 1 pharmaceuticals-15-00622-f001:**
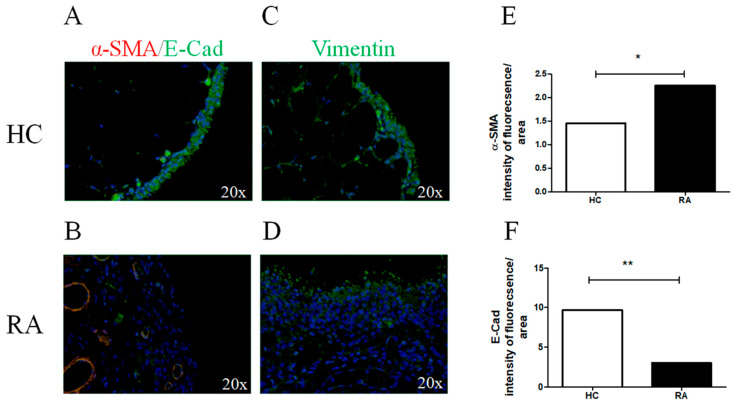
Myofibroblast markers expression in synovial tissue of patients with RA and HCs. (**A**,**B**) α-SMA (red) and E-Cad (green) immunofluorescence (IF) of HC (**A**) and RA (**B**) synovial tissue; (**C**,**D**) vimentin (green) IF of HC (**C**) and RA (**D**) synovial tissue. In RA synovial tissue, the intensity of fluorescence of α-SMA (**E**) was increased, whereas E-Cad (**F**) was decreased when compared to HCs. The histogram showed median for each synovial tissue (* = *p* = 0.01; ** = *p* = 0.001). Original magnification ×20.

**Figure 2 pharmaceuticals-15-00622-f002:**
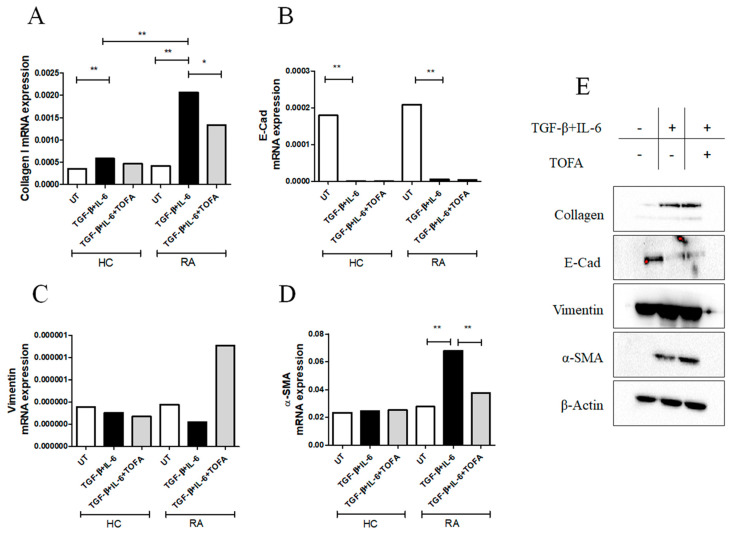
The effects of TGF-β and IL-6 with or without tofacitinib (TOFA) stimulations on RA-FLSs. (**A**) qRT-PCR of collagen I (TGF-β + IL-6 stimulation: collagen I mRNA expression in HCFLSs treated with TGF-β + IL-6 0.00058 (0.00047–0.00072) vs. collagen I mRNA expression in untreated (UT)-HC-FLSs 0.00035 (0.00024–0.00041), *p* = 0.002; collagen I mRNA expression in RA-FLSs treated with TGF-β + IL-6 0.0020 (0.0018–0.0035) vs. collagen I mRNA expression in RA-HC-FLSs 0.00041 (0.00027–0.0013), *p* = 0.002), (TOFA stimulation: collagen I mRNA expression in RA-FLSs treated with TGF-β + IL-6 0.0020 (0.0018–0.0035) vs. collagen I mRNA expression in RA-FLSs treated with TGF-β + IL-6 and TOFA 0.0013 (0.00029–0.0020), *p* = 0.01). (**B**) E-Cad (TGF-β + IL-6 stimulation: E-Cad mRNA expression in HC-FLSs treated with TGF-β and IL-6 8.38 × 10^−7^ (4.23 × 10^−7^–1.46 × 10^−6^) vs. E-Cad mRNA expression in UT-HC-FLSs 0.00017 (7.69 × 10^−5^–0.00023), *p* = 0.002; E-Cad mRNA expression in RA-FLSs treated with TGF-β + IL-6 5.99 × 10^−6^ (7.67 × 10^−7^–7.70 × 10^−6^) vs. E-Cad mRNA expression in UT-RAFLSs 0.00020 (1.28 × 10^−5^–0.00030), *p* = 0.002). (**C**) Vimentin, no significant results were retrieved following stimulation with TGF-β + IL-6 with or without TOFA. (**D**) α-SMA (TGF-β + IL-6 stimulation: α-SMA mRNA expression in RA-FLSs treated with TGF-β + IL-6 0.067 (0.048–0.10) vs. α-SMA mRNA expression in UT-RA-FLSs 0.027 (0.01–0.043), *p* = 0.002); (TOFA stimulation: α-SMA mRNA expression in RA-FLSs treated with TGF-β + IL-6 0.067 (0.048–0.10) vs. α-SMA mRNA expression in RA-FLSs treated with TGF-β + IL-6 and TOFA 0.037 (0.0057–0.055), *p* = 0.004). The histogram showed median of triplicate experiments (* = *p* = 0.01; ** = *p* < 0.004). (**E**) Western blot analyses of collagen I, E-Cad, vimentin, and α-SMA; pictures are representative of all the experiments. The grouping of gels/blots cropped from different parts of the same gel. No high-contrast (overexposure) of blots was performed.

**Figure 3 pharmaceuticals-15-00622-f003:**
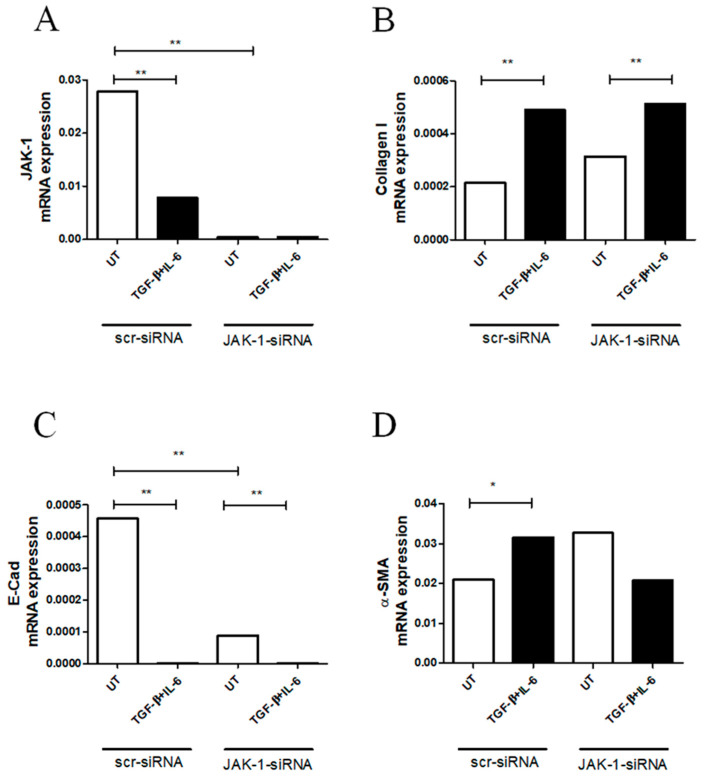
The silencing of JAK-1 gene did not fully affect the effects induced by TGF-β and IL-6. (**A**) JAK-1-siRNA efficiently knocked down JAK-1 molecule in RA-FLSs, and, after silencing, TGF-β + IL6 were unable to modulate the JAK-1 expression; qRT-PCR of JAK- 1 (JAK-1 mRNA expression in scr-siRNA untreated (UT)-RA-FLSs 0.027 (0.023–0.042) vs. JAK-1 mRNA expression in JAK-1-siRNA UT-RA-FLSs 0.00037 (0.00035–0.00050, *p* = 0.002), (JAK-1 mRNA expression in scr-siRNA UT-RA-FLSs 0.027 (0.023–0.042) vs. JAK-1 mRNA expression in scr-siRNA RA-FLSs treated with TGF-β + IL-6 0.0078 (0.0029–0.0093), *p* = 0.002). (**B**) In the RA-FLSs silenced with JAK-1-siRNA, TGF-β + IL-6 stimulation induced a more significant increase in collagen I than UT cells; collagen I (collagen I mRNA expression in JAK-1-siRNA UT-RA-FLSs 0.00031 (0.00030–0.00040) vs. collagen I mRNA expression in JAK-1-siRNA RA-FLSs treated with TGF-β + IL-6 0.00051 (0.00043–0.0025), *p* = 0.002; Collagen I mRNA expression in scr-siRNA UT-RA-FLSs 0.00021 (0.00013–0.00026) vs. collagen I mRNA expression in scr-siRNA RA-FLSs treated with TGF-β + IL-6 0.00048 (0.00042–0.00062), *p* = 0.002). (**C**) In the cells silenced with JAK-1-siRNA, the treatment with TGF-β + IL-6 promoted a significant decrease in E-Cad mRNA expression when compared to UT cells; E-Cad, (ECad mRNA expression in JAK-1-siRNA UT-RA-FLSs 8.98 × 10^−5^ (7.50 × 10^−5^–0.0001512) vs. E-Cad mRNA expression in JAK-1-siRNA RA-FLSs treated with TGF-β + IL-6 9.45 × 10^−7^ (2.02 × 10^−8^–6.52 × 10^−6^), *p* = 0.004; E-Cad mRNA expression in scr-siRNA UT-RA-FLSs 0.00045 (0.00040–0.00094) vs. E-Cad mRNA expression in scr-siRNA RA-FLSs treated with TGF-β + IL-6 7.03 × 10^−7^ (3.07 × 10^−9^–2.12 × 10^−6^), *p* = 0.004). (**D**) In scr-siRNA, TGF-β and IL-6 induced a significant increase in α-SMA mRNA levels. α-SMA; (α-SMA mRNA expression in scr-siRNA UT-RA-FLSs 0.020 (0.020–0.03) vs. α-SMA mRNA expression in scr-siRNA RA-FLSs treated with TGF-β + IL-6 0.036 (0.027–0.041), *p* = 0.04). The histogram showed median of triplicate experiments (* = *p* < 0.05 > 0.001; ** = *p* < 0.001).

## Data Availability

All the generated data are included in the body of the article or uploaded as Appendix A.

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
