# Peer review of "Tofacitinib May Inhibit Myofibroblast Differentiation from Rheumatoid-Fibroblast-like Synoviocytes Induced by TGF-β and IL-6"

_pharmaceuticals, 2022, doi:10.3390/ph15050622_

Round 1

Reviewer 1 Report

Very interesting piece of work, well designed and conducted. The text is clear and correctly presented.

The brief report should be accepted after necessary minor revisions, as herein described:

Abstract: The abstract should be a single paragraph and should follow the style of structured abstracts, but without headings. Please, remove the headings. Instead of a background short description you have stated the aim of the study (already at the end of the introduction section). Please, rephrase the background part of the abstract.

Results: Figures should be inserted into the main text close to their first citation in order to facilitate data reading.

Figures’ captions: Please format captions text removing space after the paragraph.

Please homogenize/correct throughout the methods text: ml or mL; separate values from units.

Supplementary data should be presented on a separate file.

Please, update the references for the required style.

Author Response

Reviewer 1

Very interesting piece of work, well designed and conducted. The text is clear and correctly presented.

The brief report should be accepted after necessary minor revisions, as herein described.

We would like to thank the Reviewer for the interest in our work, we revised our manuscript according to the raised queries.

Abstract: The abstract should be a single paragraph and should follow the style of structured abstracts, but without headings. Please, remove the headings. Instead of a background short description you have stated the aim of the study (already at the end of the introduction section). Please, rephrase the background part of the abstract.

Following the Reviewer’s query, we rephrased the abstract.

Results: Figures should be inserted into the main text close to their first citation in order to facilitate data reading.

Corrected, as suggested.

Figures’ captions: Please format captions text removing space after the paragraph.

Corrected, as suggested.

Please homogenize/correct throughout the methods text: ml or mL; separate values from units.

Corrected, as suggested.

Supplementary data should be presented on a separate file.

We uploaded supplementary data as a separate file.

Please, update the references for the required style.

We updated the references according to the Journal style and the suggestion of Reviewer 3.

Reviewer 2 Report

The paper analyzes TOFA effects  on the myofibroblast. 

Line 87.  Authors write: The effects of TGF-β+IL-6 on Collagen I expression in RA-FLSs were significantly higher than HCs-FLSs.
This effects is natural according to IL-6 pathway induction in RA-FLSs. Why JAK and STATs expression were not studied  in TOFA treated cells? Further, STAT expression is not indicator of JAK/STAT pathway activation. The  phosphorylation of the factors should be analyzed.

Line 182. more agressive phenotype- needs to be explained.

Line 199.  Authors write: Considering the role of JAK/STAT pathway in this context [10,11], we assessed the possible inhibiting role of tofacitinib. However, I can not find the results for JAK/STAT expression/phosphorylation in the TOFA treated group.

 4.5. FLSs isolation and culture
Flow cytometry analysis description is not complet.
Please indicate gating strategy of homogeneous population (phenotype: 0% CD14, >90% CD90 and 7% CD68+/CD90+) obtained. Please characterize the antibodies used for the cell staining .

Line 305. What 199 medium means?

Line 332, 204 Authors indicated that  tofacitinib reduced the expression of both Collagen I and α-SMA on RA-FLSs, however  Supplementary Figure 2 indicate that  (TOFA) administration was unable to prevent TGF-β+IL-6 effects. Please verify the conclusions. For which group are the results of Supplementary Figure 2?

Author Response

Reviewer 2

The paper analyzes TOFA effects on the myofibroblast. 

We would like to thank the Reviewer for the interest in our work, we revised our manuscript according to the raised queries.

Line 87.  Authors write: The effects of TGF-β+IL-6 on Collagen I expression in RA-FLSs were significantly higher than HCs-FLSs. This effects is natural according to IL-6 pathway induction in RA-FLSs. Why JAK and STATs expression were not studied in TOFA treated cells? Further, STAT expression is not indicator of JAK/STAT pathway activation. The  phosphorylation of the factors should be analyzed.

We would like to thank the Reviewer for this remark allowing us to better specify our study rationale. As rightly stated by the Reviewer, the phosphorylation of JAK/STAT factors would be the best indicator of the activity of this pathway. However, the inhibition of phosphorylation of JAK/STAT factors on synovial cells as well as immune cells following the treatment with tofacitinib has been already demonstrated. These data are already available in literature [Boyle DL, et al. Ann Rheum Dis. 2015;74:1311-6; Palmroth M, et al. Front Immunol. 2021;12:738481;

Isailovic N et al. Clin Exp Immunol 2021;205:142-9]. Thus, we did not replicate these experiments but we focused on additional possible mechanisms of tofacitinib in RA-FLSs exploring further effects of this drug on markers of fibroblast activation. We better specified these findings in our study adding a paragraph in the discussion about that. We also pointed out the hypothesis-generating nature of our study to be fully confirmed in subsequent research. Considering this point, we also tempered the discussion about our results and the conclusions.

Line 182. more agressive phenotype- needs to be explained.

Many thanks for this point. We changed “more aggressive phenotype” with a more appropriate “activated phenotype”.

Line 199.  Authors write: Considering the role of JAK/STAT pathway in this context [10,11], we assessed the possible inhibiting role of tofacitinib. However, I cannot find the results for JAK/STAT expression/phosphorylation in the TOFA treated group.

Similarly to previous query, we better specified that the inhibition of phosphorylation of JAK/STAT factors on synovial cells as well as on immune cells following the treatment with tofacitinib has been already demonstrated. These data are already available in literature [Boyle DL, et al. Ann Rheum Dis. 2015;74:1311-6; Palmroth M, et al. Front Immunol. 2021;12:738481]. Thus, we did not replicate these experiments but we focused on additional possible mechanisms of tofacitinib in RA-FLSs exploring additional effect of this drug on markers of fibroblast activation. Considering this point, we rephrased our manuscript accordingly.

 4.5. FLSs isolation and culture

Flow cytometry analysis description is not complet.

Please indicate gating strategy of homogeneous population (phenotype: 0% CD14, >90% CD90 and 7% CD68+/CD90+) obtained. Please characterize the antibodies used for the cell staining.

As suggested, we better detailed the methods of flow cytometry analysis and we added the antibodies which we used for cell staining.

Line 305. What 199 medium means?

Apologies for the lack of clarity, this would be a mistake which we corrected.

Line 332, 204 Authors indicated that tofacitinib reduced the expression of both Collagen I and α-SMA on RA-FLSs, however Supplementary Figure 2 indicate that  (TOFA) administration was unable to prevent TGF-β+IL-6 effects. Please verify the conclusions. For which group are the results of Supplementary Figure 2?

Many thanks for this remark, we tempered our conclusion according to our results. We specified that the results of Supplementary figure 2 are referred to RA-FLSs.

Reviewer 3 Report

Dear Authors,

Please improve your introduction 

Please add more references

Please add more conclusions

Kind regards

Author Response

Reviewer 3

Dear Authors, Please improve your introduction; Please add more references; Please add more conclusions.

We would like to thank the Reviewer for the interest in our work. As suggested, we improved the introduction, added more references, and implemented the conclusions.

Reviewer 4 Report

EMT of RA synovial fibroblasts was little known.There where few studies about EMT mechanism in RA, not to mention the influence of drug. The study provided several interesting insights about this previously untouched field.

Author Response

Reviewer 4

EMT of RA synovial fibroblasts was little known. There where few studies about EMT mechanism in RA, not to mention the influence of drug. The study provided several interesting insights about this previously untouched field.

We would like to thank the Reviewer for the interest in our work, no queries to be addressed.